

# A scoping review of determinants of performance in dressage

Sarah Jane Hobbs[1], Lindsay St George[1], Janet Reed[1], Rachel Stockley[1], Clare Thetford[1], Jonathan Sinclair[1], Jane Williams[2], Kathryn Nankervis[2] and Hilary M. Clayton[3]

[1] University of Central Lancashire, Preston, United Kingdom
[2] Hartpury University, Gloucester, United Kingdom
[3] Sport Horse Science, Mason, MI, United States of America

Corresponding author
Sarah Jane Hobbs,
sjhobbs1@uclan.ac.uk

## ABSTRACT

As a first step in achieving an evidence-based classification system for the sport of Para Dressage, there is a clear need to define elite dressage performance. Previous studies have attempted to quantify performance with able-bodied riders using scientific methods; however, definitive measures have yet to be established for the horse and/or the rider. This may be, in part, due to the variety of movements and gaits that are found within a dressage test and also due to the complexity of the horse-rider partnership. The aim of this review is therefore to identify objective measurements of horse performance in dressage and the functional abilities of the rider that may influence them to achieve higher scores. Five databases (SportDiscuss, CINAHL, MEDLINE, EMBASE, VetMed) were systematically searched from 1980 to May 2018. Studies were included if they fulfilled the following criteria: (1) English language; (2) employ objective, quantitative outcome measures for describing equine and human performance in dressage; (3) describe objective measures of superior horse performance using between-subject comparisons and/or relating outcome measures to competitive scoring methods; (4) describe demands of dressage using objective physiological and/or biomechanical measures from human athletes and/or how these demands are translated into superior performance. In total, 773 articles were identified. Title and abstract screening resulted in 155 articles that met the eligibility criteria, 97 were excluded during the full screening of articles, leaving 58 included articles (14 horse, 44 rider) involving 311 equine and 584 able-bodied human participants. Mean $\pm$ sd (%) quality scores were $63.5 \pm 15.3$ and $72.7 \pm 14.7$ for the equine and human articles respectively. Significant objective measures of horse performance ($n = 12$ articles) were grouped into themes and separated by gait/movement. A range of temporal variables that indicated superior performance were found in all gaits/movements. For the rider, $n = 5$ articles reported variables that identified significant differences in skill level, which included the postural position and ROM of the rider's pelvis, trunk, knee and head. The timing of rider pelvic and trunk motion in relation to the movement of the horse emerged as an important indicator of rider influence. As temporal variables in the horse are consistently linked to superior performance it could be surmised that better overall dressage performance requires minimal disruption from the rider whilst the horse maintains a specific gait/movement. Achieving the gait/movement in the first place depends upon the intrinsic characteristics of the horse, the level of training achieved and the ability of the rider to apply the correct aid. The information from this model

will be used to develop an empirical study to test the relative strength of association between impairment and performance in able-bodied and Para Dressage riders.

# INTRODUCTION

Para-equestrian Dressage is an internationally-recognized sport that provides educational and competitive opportunities in the sport of dressage for athletes with impairments. In para-equestrian competition, the rider is classified according to his or her functional ability and, based on this classification, competitors are grouped to ensure a level playing field. To improve the validity of classification across Paralympic sports, the International Paralympic Committee (IPC) mandates the development of evidence-based classification systems, in order to minimise the impact of impairment on competition outcomes by grouping athletes in Sport Classes based on the extent of activity limitation caused (*Tweedy & Vanlandewijck, 2011*). To accomplish this, all sport-specific classification systems must develop scientific evidence to define: eligible types of impairment, eligible impairment severity and the extent of activity limitation caused (*Tweedy & Vanlandewijck, 2011*). To determine the extent of activity limitation, the relative strength of association between impairment and fundamental sport-specific activities/skills, or ''performance measures'', must be determined (*Tweedy & Vanlandewijck, 2011*; *Tweedy, Beckman & Connick, 2014*; *Tweedy, Mann & Vanlandewijck, 2016*). Therefore, performance measures are determined by a comprehensive understanding of fundamental skills, abilities and body positions required for sport-specific performance (*Tweedy, Mann & Vanlandewijck, 2016*). As a first step in achieving an evidence-based classification system for the sport of Para Dressage, there is a clear need to define performance measures by reviewing the scientific literature to determine objective measurements of the athletes (rider and horse) that predict elite dressage performance.

Performance in dressage is measured by a percentage score that is awarded by judges in relation to a pre-defined test performed by a horse-rider combination. The test consists of a series of gaits and patterns with each segment receiving a separate score. The evaluation and resulting scores will be based on the Scale of Training as described for example by *British Dressage (2019)*, see Table S1. Previous studies have attempted to quantify the determinants of elite riding/dressage performance using able-bodied riders and various kinematic techniques, however definitive measures have yet to be established for the horse and/or the rider. This may be, in part, due to the variety of movements and gaits that are found within a single test and also due to the complexity of the horse-rider partnership.

When measuring horse performance, some studies have only focused on diagonal gaits (*Back et al., 1994*; *Clayton, 1994a*; *Holmstrom, Fredricson & Drevemo, 1994a*; *Holmstrom & Drevemo, 1997*; *Morales et al., 1998*) others on walk (*Clayton, 1995*; *Back, Schamhardt & Barneveld, 1996*) or gallop gaits (*Deuel & Park, 1990a*; *Clayton, 1994b*), and some

on transitions between gaits (*Argue & Clayton, 1993a*; *Argue & Clayton, 1993b*; *Tans, Nauwelaerts & Clayton, 2009*). In these studies, good horse performance is usually determined by selecting successful high-level performers, by comparing between two different groups of horses, or by comparing measurements to judged scores. The majority of studies are limited to straight-line motion, particularly those using two-dimensional video-based techniques, although a small number have obtained measurements during a prescribed dressage test (*Deuel & Park, 1990a*; *Deuel & Park, 1990b*; *Biau & Barrey, 2004*). Kinematic measures that have been identified as important performance determinants include temporal variables such as hind first diagonal dissociation (*Holmström, Fredricson & Drevemo, 1994b*; *Clayton, 1997*; *Hobbs, Bertram & Clayton, 2016*), joint range of motion (ROM) such as increased forelimb fetlock extension (*Back et al., 1994*), and centre of mass (COM) motion such as increased dorso-ventral displacement of the trunk (*Biau & Barrey, 2004*). Although some measures consistently define better performance, others are also dependent on the gait or movement being performed, which adds to the complexity of defining overall performance.

The performance of the rider alone (which relates directly to rider skill and accuracy) carries few marks in dressage, so performance from a rider perspective relates mainly to their ability to influence the horse's performance. The rider's position has been described (*Lovett, Hodson-Tole & Nankervis, 2005*), evaluated in relation to skill level (*Schils et al., 1993*; *Kang et al., 2010*) and related to the presence of rider asymmetries (*Symes & Ellis, 2009*; *Gandy et al., 2014*; *Alexander et al., 2015*). Few studies have investigated the rider's influence on the horse and, of these, the majority have focused on the phasic relationships between horse and rider motion, based on the principle that movements of an experienced rider are more closely synchronized with the horse (*Peham et al., 2001*; *Lagarde et al., 2005*; *Münz, Eckardt & Witte, 2014*; *Baillet et al., 2017*). Studies have investigated the physiological demands of riding (*Westerling, 1983*; *Devienne & Guezennec, 2000*; *Meyers, 2006*; *De Barros Souza et al., 2008*; *Roberts, Shearman & Marlin, 2009*; *Beale et al., 2015*; *Sung et al., 2015*; *Sainas et al., 2016*; *Baillet et al., 2017*) and others have evaluated cardiovascular fitness and other aspects of rider fitness (*Westerling, 1983*; *Devienne & Guezennec, 2000*; *Meyers & Sterling, 2000*; *Meyers, 2006*; *Beale et al., 2015*; *Sung et al., 2015*; *Sainas et al., 2016*).

As a first step toward identifying determinants of performance for the rider, this study will review the scientific literature defining (a) locomotion patterns of the horse that quantify gait quality, (b) rider demands, (c) rider functional skills and abilities, and (d) superior performance characteristics of the horse-rider dyad. From the information extracted, a theoretical model will be developed to link objective measures from the rider that may influence overall performance in dressage. The aim of this review is therefore to identify objective measurements of horse performance in dressage and the functional skills and abilities of the rider that may influence them to achieve higher scores. The review is part of a larger project commissioned by the Fédération Equestre Internationale. Objective measures of performance identified in the review will be used to evaluate performance and the effect of impairment on performance in experimental studies of able-bodied and para dressage riders at a later date.

## SURVEY METHODOLOGY

### Search strategy

A review framework (Table S2) was developed, detailing study population, outcomes and setting required to fulfil the review objectives. A search strategy (Table S3) was then developed and included the following general keywords: "horse riding", "elite dressage", "paradressage or para-dressage or para dressage", "dressage performance", horseback riding". As the review was designed to be a broad systematic search, additional keywords were developed and grouped according to the following overarching themes: rider, horse, rider physiology, rider psychology, impaired rider performance, rider performance, horse performance, horse training and outcome measures. Overarching themes were combined with the general keywords to develop the search strategy. The following databases were systematically searched from 1980 to May 2018 to identify studies for potential inclusion within the study: SportDiscuss, CINAHL, MEDLINE, EMBASE, VetMed. The searches were performed between 23rd April 2018 and 23rd May 2018.

### Study selection

Studies were included if they fulfilled the following inclusion criteria: (1) English language, (2) employed objective, quantitative outcome measures for describing rider and/or horse performance in dressage (e.g., optical motion capture/kinematic data), (3) described objective measures of superior horse performance using between-subject comparisons and/or relating outcome measures to competitive scoring methods (e.g., comparison of linear kinematic variables in elite vs. non-elite horses or the relationship between these variables and judged dressage scores), (4) described demands of dressage using objective physiological and/or biomechanical measures from riders and/or how these demands are translated into superior performance (e.g., descriptive studies on objective variables like heart rate, muscle strength, or joint ROM in one rider cohort or a comparison of these variables across varying levels of riders to describe superior performance). Only full-text, peer-reviewed scientific articles were included. Conference proceedings/abstracts, theses and grey literature were excluded. Title and abstract screening was conducted by two reviewers (LSG, EL) to determine whether each study met the inclusion criteria and any inconsistencies were settled through discussion with a third reviewer (SJH).

The search produced 679 results. Hand searches were conducted by reviewing reference lists of studies meeting the inclusion criteria and relevant review articles (*Hall et al., 2008*; *Douglas, Price & Peters, 2012*; *Janura et al., 2012*; *Clayton & Hobbs, 2017a*; *Hall & Heleski, 2017*), producing an additional 94 articles that were not previously identified. The study selection is presented as a PRISMA flow diagram (*Moher et al., 2009*) in Fig. 1.

### Data extraction and synthesis

Included articles underwent data extraction and methodological quality assessment, which was independently conducted by two reviewers (LSG, EL) using the "Critical Review Form for Quantitative Studies" (*Law et al., 1998*). In accordance with *Zadnikar & Kastrin (2011)*, methodological quality was assessed using 16 dichotomous items that were scored as either 1 or 0 for studies that fulfilled or did not fulfil each criterion, respectively (horse, Table S4;

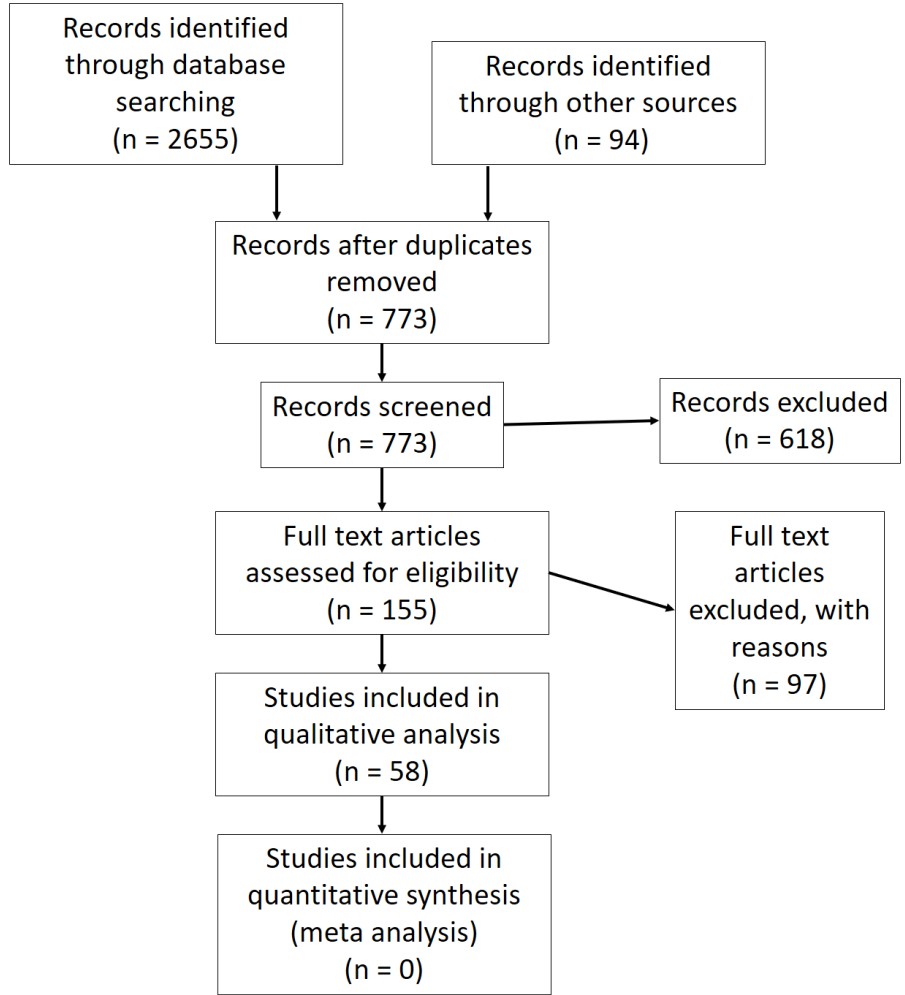

**Figure 1** PRISMA diagram illustrating the study selection process.

rider, Table S5). Intervention description, validity and reliability of outcome measures and the clinical relevance of significant differences between groups formed 4 of the 16 items for quality scoring but were not relevant for the majority of study designs included in this review. Thus, "not applicable" scores were given where appropriate for individual studies and the score for each article was calculated as a percentage of the total scores given. A score >80% was considered to indicate excellent methodological quality. Each reviewer independently extracted the following information from each article: citation, study purpose, study design, sample size and description, outcome measures, main results and main conclusion. Data extraction forms and methodological quality scores were compared between both reviewers and inconsistencies or disagreements were resolved by discussion and consensus.

The heterogeneity of the included studies prohibited pooling of data for meta-analysis and overall effect size calculations. Thus, descriptive summary tables were used to synthesize results for horse (Table S6) and rider (Table S7) performance measures. Summary tables

include the extracted data and quality score for each included study. Separate summary tables were also created to group significant objective outcome measures, employed by the included studies, based on overarching themes related to horse (Table S8) and rider (Table S9) performance. For the horse, these overarching themes include: temporal measures, joint/segment kinematics, trunk motion, impulsion, stride length/adjustability (linear kinematics) and connection. Adjustability refers to the ease with which the horse shortens and lengthens the stride. Connection refers to the harmonious interaction between rider and horse revealed in the generation of energy that is controlled by the contact of the rider's hand with the horse's mouth via the rein. Outcome measures for horse performance were only included in the summary table if selected studies had described a direct relationship between the outcome measure and performance (for example: better competition scores or a difference between elite and non-elite horses). For the rider, these overarching themes included: range of motion (ROM), strength, rider fitness, physiological demands of riding, coordination between horse and rider, rider coordination and balance. Within each overarching theme, outcome measures for both horse and rider were further grouped based on the measurement tool (for example: accelerometer, kinematics), the gait or movement from which the measure was obtained and the studies that employed this measure. Where available in the literature, mean ± standard deviation (sd) data were included for rider outcome measures, and where multiple studies reported values for the same outcome measure, overall mean ± sd values were calculated (see Table S9). A thematic network (based on *Attride-Stirling (2001)*) was created to illustrate objective measures of dressage horse performance (global theme) from the literature, with overarching themes and their objective performance measures representing the basic themes and the associated gaits/movements representing the organising themes (Fig. 2, Table 1). For the rider, significant performance outcome measures associated with a gait were used to develop a theoretical model to link the potential influence of rider skill to horse performance (see Fig. 3, Table 2, Text S1).

## RESULTS

Title and abstract screening resulted in 155 articles that met the eligibility criteria. Ninety-seven articles were excluded during the full screening of articles, resulting in 58 included articles (14 for horse, 44 for rider) involving 311 equine and 584 able-bodied human participants. Mean ± sd (%) quality scores were 63.5 ± 15.3 and 72.7 ± 14.7, for horse and rider performance respectively.

### Horse performance

Significant objective measures of performance, grouped according to horse overarching themes, including method of measurement, gait or type of movement and performance effects are shown in Table S8. A thematic network linking significant outcome measures with gaits/movements, classified using colour for overarching themes summarizes this information in Fig. 2.

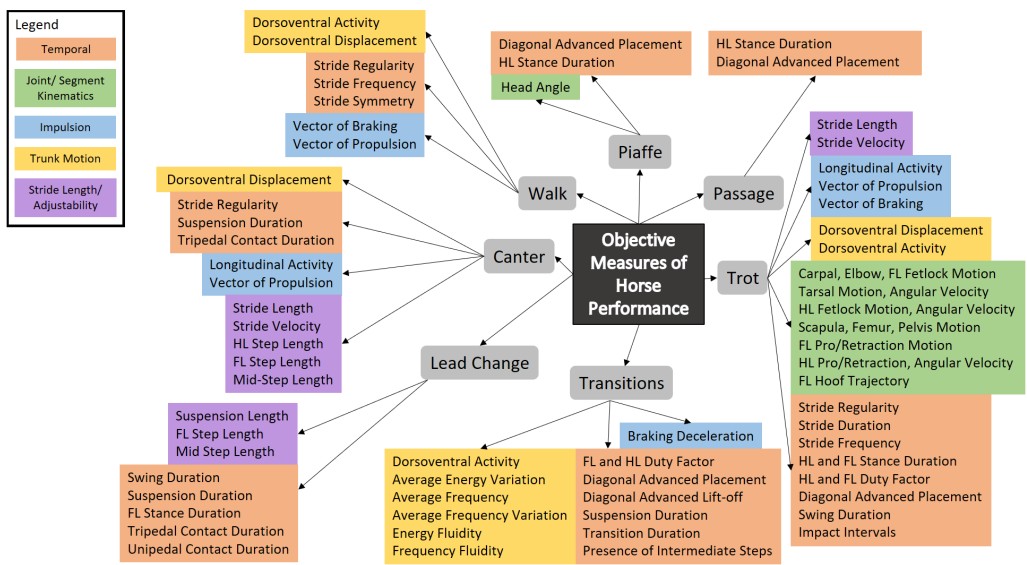

**Figure 2** Objective measures of horse performance grouped by gait and overarching theme.

**Table 1** Articles included in the thematic network of objective horse performance measures and their respective quality scores.

| Articles | Gaits/Movements | Theme(s) | Quality score |
|---|---|---|---|
| *Argue & Clayton (1993a)* | Transitions | Temporal | 53.3 |
| *Back et al. (1994)* | Trot | Temporal, Joint/Segment kinematics | 66.7 |
| *Biau, Lemaire & Barrey (2002)* | Transitions | Temporal, Trunk motion, Impulsion | 46.7 |
| *Biau & Barrey (2004)* | Walk, trot, canter | Temporal, Trunk motion, Impulsion | 42.9 |
| *Clayton (1997)* | Trot, passage, piaffe | Temporal | 85.7 |
| *Deuel & Park (1990a)* | Extended canter, canter lead changes (included in transitions) | Temporal, Stride length/Adjustability | 57.1 |
| *Deuel & Park (1990b)* | Extended trot | Temporal, Stride length/Adjustability | 57.1 |
| *Holmstrom, Fredricson & Drevemo (1994a)* | Trot | Temporal, Joint/Segment kinematics | 66.7 |
| *Holmstrom & Drevemo (1997)* | Trot | Joint/Segment kinematics | 53.3 |
| *Tans, Nauwelaerts & Clayton (2009)* | Transitions | Temporal | 66.7 |
| *Lashley et al. (2014)* | Piaffe | Joint/Segment kinematics | 80.0 |
| *Morales et al. (1998)* | In-hand trot | Joint/Segment kinematics | 73.3 |

## Rider performance

Significant outcome measures, method of measurement, gait or type of movement and differences between skilled riders and non-skilled riders or non-riders are reported in Table S9.
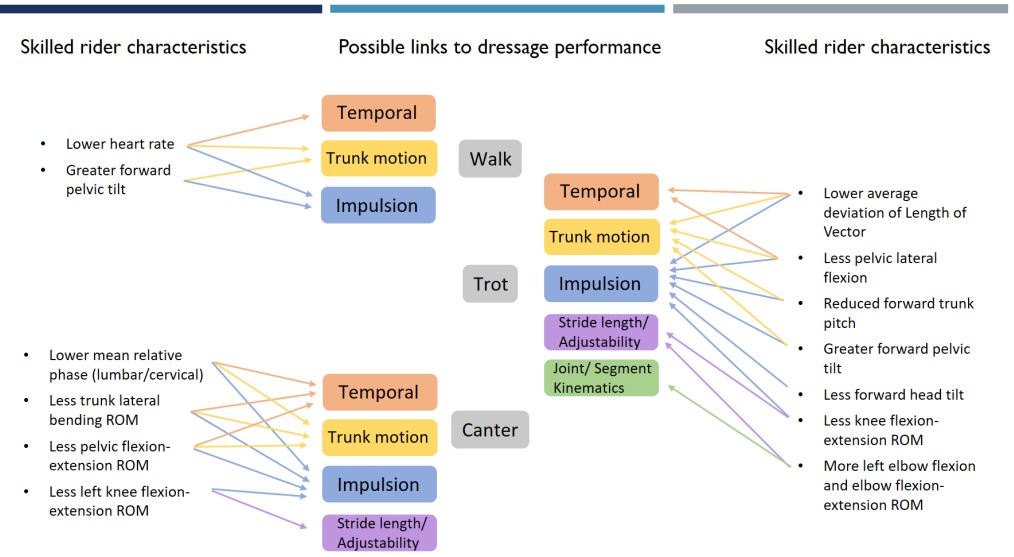

**Figure 3 Objective measures of skilled rider characteristics grouped by gait with arrows showing theoretical links to horse performance themes.** Theoretical links between rider skill and horse performance themes are developed using evidence presented within the discussion. Extracts from the discussion are provided in Text S1 to highlight the reasoning behind these links.

**Table 2 Articles included in the theoretical model linking skilled rider characteristics to horse performance themes and their respective quality scores.**

| Article | Gaits/Movements | Skilled rider characteristics | Quality score |
|---|---|---|---|
| Eckardt & Witte (2016) | Sitting Trot | Reduced forward trunk pitch (max $p = 0.026$, mean $p = 0.04$). Reduced knee flexion-extension ROM ($p < 0.01$). Reduced forward head tilt ($p = 0.04$). Greater flexion-extension ROM for left elbow ($p = 0.02$) and greater elbow flexion ($p < 0.01$). | 66.7 |
| | Canter | Reduced trunk lateral bending ROM ($p < 0.05$). Reduced left knee flexion-extension ROM ($p < 0.05$). | |
| Münz, Eckardt & Witte (2014) | Walk | Greater forward pelvic tilt ($p < 0.05$). | 78.6 |
| | Sitting Trot | Greater forward pelvic tilt ($p < 0.05$). Reduced pelvic lateral flexion ($p < 0.05$). | |
| | Canter | Reduced pelvic flexion-extension ROM ($p < 0.05$). | |
| Olivier et al. (2017) | Simulated gallop (included in canter) | Lower mean relative phase (lumbar/cervical) ($p = 0.009$). | 86.7 |
| Peham et al. (2001) | Sitting Trot | Lower average deviation of length of vector ($p < 0.05$). | 73.3 |
| Sung et al. (2015) | Walk | Lower heart rate ($p = 0.021$). | 56.3 |

## Linking rider and horse performance

A schematic diagram of a proposed theoretical model linking rider skills to horse performance is provided in Fig. 3, based on significant outcome measures identified for the horse and the rider.

## DISCUSSION

This review was designed to (a) identify objective measurements of horse performance that would be awarded higher scores in a dressage test, (b) identify functional skills and abilities of higher level riders and (c) predict the association between rider skill/ability on horse performance in dressage. Previous reviews of the literature have described the physiological and/or biomechanical traits of riders (*Douglas, Price & Peters, 2012*; *Clayton & Hobbs, 2017a*), with *Clayton & Hobbs (2017a)* describing the biomechanical horse-rider interaction during walk, trot, canter and gallop. However, no previous reviews have considered both physiological and biomechanical abilities and/or skills of the rider and horse to establish what factors are most important for predicting superior riding performance. Thus, this review presents an original summary of fundamental rider and horse skills/abilities and the influence of these on the horse-rider partnership, which will be used in the future to define "performance measures" for developing an evidence-based Classification System for Para Dressage.

Intrinsic factors of the horse will influence performance and judged scores and so horse selection is an important consideration with regards the horse's potential to perform well in dressage (*Back et al., 1994*; *Holmström, Fredricson & Drevemo, 1994b*). The subsequent training determines whether the horse reaches its full athletic potential and the skill of the rider is key to both maximizing the benefits of training and to producing a top performance in the competition arena. Qualities of the horse's movement that may be influenced by the rider (dependent on the rider's ability) include stride length, stride frequency, rhythm, connection, impulsion, straightness, collection and balance (*Peham et al., 2004*; *Bradshaw et al., 2005*; *Schöllhorn et al., 2006*; *Roepstorff et al., 2009*; *De Cocq et al., 2010a*; *De Cocq et al., 2010b*; *Byström et al., 2015*; *Eiersiö et al., 2015*; *Engell et al., 2016*). However, riders differ greatly in their equitation skills, their sensitivity in assessing the horse's performance, and their competency in improving that performance, whilst horses differ in their trainability to improve their performance (*McCall, 1990*). A basic requirement of a competent dressage rider is to have an independent seat, which implies that movements of the rider's pelvis follow and compensate for the horse's trunk movements allowing the arms and legs to act independently to follow the horse's head and neck motion and to give aids to the horse.

The following discussion will explore the identified measures of horse performance as they relate to both intrinsic traits and rider controllable factors. The rider's influence will then be discussed based on significant performance measures, across gaits for which information is available.

### Walk

Walk is a symmetrical gait with a lateral sequence of footfalls. The limb support sequences alternate between bipedal and tripedal support and there are no suspension phases (*Clayton, 1995*; *Hodson, Clayton & Lanovaz, 1999*). The rules for dressage (*Fédération Equestre Internationale, 2019*) state that there should be a regular rhythm with equal intervals between footfalls and, specifically, walking with lateral couplets is described as a fault. In accordance with these rules, *Biau & Barrey (2004)* found that stride regularity was an objective measure of performance. This implies that step durations occur at intervals of
approximately 25% of stride duration which means that, at any time, only one limb provides braking and one limb provides propulsion (*Merkens & Schamhardt, 1988*). However, this is associated with out-of-phase fluctuations between potential energy and kinetic energy in the hind limbs and forelimbs which is energetically inefficient (*Griffin, Main & Farley, 2004*). However, it should be noted that energetic efficiency is not a requirement of dressage; in fact, horses are rewarded for performing with great energy and impulsion. The fact that many dressage horses show an irregular rhythm, especially in the extended walk (*Clayton, 1995*), may represent a more energetically efficient pattern of limb coordination than having a regular rhythm (*Clayton & Hobbs, 2019*). Related to this, the vector of braking and vector of propulsion were correlated to movement marks in young horses (*Biau & Barrey, 2004*), indicating that higher marks were given for good, symmetrical braking and propulsive work.

Other temporal variables identified by *Biau & Barrey (2004)* as objective measures of performance are stride symmetry and stride frequency. Stride frequency is largely determined by limb length, with taller horses having slower strides than smaller horses walking at the same speed, because the body mass moves further forward over the grounded hoof during each step (*Back et al., 1995*). The rider can influence stride frequency, but care must be taken not to induce a change in rhythm at the same time (*Wolframm, Bosga & Meulenbroek, 2013*). Horses usually walk symmetrically unless they are unloading a lame limb (*Buchner et al., 1995*) or show marked sidedness (*Byström et al., 2018*), although rider asymmetry, such as differences in rein tension, can also influence horse symmetry (*Terada, Clayton & Kato, 2006*; *Kuhnke et al., 2010*; *Eisersiö et al., 2015*).

Dorsoventral activity and dorsoventral displacement of the trunk are indicative of vertical trunk motion/acceleration. In fact, in the absence of suspension phases, vertical excursions of the trunk are smaller at walk than in the other gaits of dressage horses and this makes it an easier gait for the rider to sit compared to trot (*Byström et al., 2010*). Since the walk does not have suspension phases, back movements and pitching rotations of the horse's trunk are driven by the limb movements (*Faber et al., 2000*; *Byström et al., 2010*). Pitching rotations of the trunk are related to the relative heights of the croup and withers. At hind hoof contact the croup is low and the trunk is in its maximal nose-up pitched position. In the first half of hind limb stance, the croup rises and the trunk pitches in a nose-down direction. The direction of rotation is reversed around the time of forelimb contact which is followed by rising of the withers as the croup descends. The cycle is repeated during the movements of the contralateral limbs. Thus, the pitch of the horse's back is maximally nose-up at hind limb contact and nose-down at forelimb contact (*Byström et al., 2010*). The saddle rotates in the same direction as the horse's back (*Von Peinen et al., 2009*; *Münz, Eckardt & Witte, 2014*) and the rider's pelvis pitches in counter-rotation to the saddle (*Byström et al., 2010*); the pelvis is maximally anteriorly rotated at hind limb contact and maximally posteriorly rotated at forelimb contact. Pelvic rotations occur twice per stride and, movements of the rider and the horse are less well synchronized than for the other gaits (*Wolframm, Bosga & Meulenbroek, 2013*). Comparing skilled riders with beginners, *Münz, Eckardt & Witte (2014)* found that the pelvis underwent anterior to posterior pelvic

motion (referred to in the paper as greater forward pelvic tilt) in skilled riders compared with beginners and this was associated with increased nose-up trunk rotation of the horse.

When comparing the physiological demands for the athlete, a significant increase in heart rate in amateur athletes during two minutes of walking compared to skilled athletes was attributed to differences in physical fitness by *Sung et al. (2015)*, as their resting heart rates were also significantly different. It is interesting to note that the heart rate increase at walk compared to resting values was 28.9% in amateur athletes and 14.1% in skilled athletes, but when practicing jumping the increases over resting values were 73.5% and 84.7%, respectively. This could indicate that skilled athletes do not work as hard during walking, due to developing better relaxation and/or harmony with the horse. *Münz, Eckardt & Witte (2014)* suggested that the pelvis of less skilled athletes moves "ahead" of the horses' movement and *Wolframm, Bosga & Meulenbroek (2013)* found lower interclass correlations between horse and rider motion in walk compared to canter. Out of phase timing of the rider with the horse may disrupt the rhythm of the horse and result in greater energy expenditure for the less skilled athlete to maintain an active walk.

## Trot

Trot is a diagonally-coordinated, symmetrical gait in which diagonal support phases alternate with suspension phases (*Clayton, 1994a*; *Holmström, Fredricson & Drevemo, 1994b*). Diagonal dissociation at the start and end of the diagonal support phases occurs frequently and gives rise to short periods of single support (*Holmström, Fredricson & Drevemo, 1994b*; *Hobbs, Bertram & Clayton, 2016*). The rules for dressage (*Fédération Equestre Internationale, 2019*) state that the trot should show free, active and regular steps with trot quality being judged by the regularity and elasticity of the steps, the cadence and impulsion (where impulsion indicates movement due to the storage and release of elastic energy in the tendinous tissues of the limbs), the suppleness of the back, the engagement of the hindquarters, and the ability to maintain the same rhythm and natural balance in all variations of the trot (collected, working, medium, extended). Thus, it is not surprising that improved stride regularity and symmetry (*Biau & Barrey, 2004*) were predictive of better performance.

The ability to increase stride length and velocity were identified as important traits (*Deuel & Park, 1990b*). At trot, velocity is increased by taking longer and/or faster strides. However, the dressage rules (*Fédération Equestre Internationale, 2019*) require that the same stride frequency be maintained regardless of speed, so changes in velocity rely on the ability to adjust stride length (*Clayton, 1994a*). The horse's inherent stride length is affected by leg length and the variable that best reflects changes in stride length is tracking length, i.e., the distance between the hoof print of a fore hoof and the following hoof print of the ipsilateral hind hoof in the direction of travel (*Clayton, 1994a*). Greater over-tracking length (where the ipsilateral hind hoof print is further forward than the ipsilateral fore hoof) and longer strides are achieved by increasing forward propulsion and the propulsion vector, indicating greater overall propulsive work, was correlated to total score in experienced horses (*Biau & Barrey, 2004*). Muscular strength determines the development of propulsive force; more powerful muscles are able to generate force more rapidly which is evident in a shorter

stance duration (*Back et al., 1994*). Shorter stance durations were identified as having a positive influence on stride quality both in the forelimbs (*Deuel & Park, 1990a*; *Deuel & Park, 1990b*) and hindlimbs (*Holmström, Fredricson & Drevemo, 1994b*) resulting in smaller duty factors (*Holmström, Fredricson & Drevemo, 1994b*). Swing durations are influenced by both stride duration and stance duration. *Deuel & Park (1990b)* found that shorter hind limb and longer forelimb swing durations were objective measures of performance. Taken together, having relatively shorter stance phases and relatively longer swing phases is aesthetically pleasing because the trot appears less grounded and more bouncy, which is regarded positively by judges and explains the greater dorsoventral displacement and activity found by *Biau & Barrey (2004)*.

Diagonal dissociation is a short temporal separation of the limbs of the diagonal pairs at contact and/or lift off. The period of dissociation is too short to be perceived by the human eye and was detected in early studies using frame-by-frame evaluation (*Clayton, 1994a*; *Holmström, Fredricson & Drevemo, 1994b*). Hind first diagonal dissociation at contact has been identified as a beneficial characteristic (*Holmström, Fredricson & Drevemo, 1994b*) and is associated with an uphill posture of the horse's trunk (*Holmström, Fredricson & Drevemo, 1994b*; *Hobbs, Bertram & Clayton, 2016*). Dissociation at lift off is usually hind first and it has been suggested that the short period of forelimb single support at the end of diagonal stance may provide a vertical force that contributes to the maintenance of uphill posture (*Hobbs, Bertram & Clayton, 2016*).

It is not surprising that limb kinematics would influence dressage scores, as most people concentrate on swing phase limb movements when evaluating ridden gaits (*Holmström, Fredricson & Drevemo, 1994b*). In trot, significant objective hind limb measurements of performance include greater pelvic inclination, larger hindlimb pendulation associated with greater hindlimb protraction, greater tarsal flexion and faster angular velocity in the hock joint (*Holmström, Fredricson & Drevemo, 1994b*; *Holmstrom & Drevemo, 1997*). In the forelimbs objective measurements include greater swing phase retraction, increased elbow and carpal flexion at the beginning of swing phase retraction (*Holmström, Fredricson & Drevemo, 1994b*), greater fetlock extension in stance (*Back et al., 1994*; *Holmström, Fredricson & Drevemo, 1994b*; *Morales et al., 1998*), greater scapular ROM (*Back et al., 1994*; *Holmström, Fredricson & Drevemo, 1994b*; *Morales et al., 1998*), and higher forelimb hoof trajectory in swing (*Holmström, Fredricson & Drevemo, 1994b*).

Some of these attributes are associated with other aspects of gait quality, some are modifiable, and some are inherent qualities of the horse. Hind-first diagonal dissociation and greater hindlimb protraction are associated with greater nose up pitching of the trunk (*Hobbs, Bertram & Clayton, 2016*). Diagonal dissociation can be modified by changing speed, but some horses maintain hind first dissociation across a larger speed range, which is a desirable trait (*Hobbs, Bertram & Clayton, 2016*). Shorter stance durations, greater fetlock extension, faster extension of the hock and rotation of the pelvis in late stance, and flexion of the hindlimb joints are all related to the ability of the horse to store and release energy and they are largely responsible for creating impulsion. The physiological condition and conformational traits of the horse will influence the horses' ability to store and release energy (*Back et al., 1994*; *Holmström, Fredricson & Drevemo,*

*1994b*; *Morales et al., 1998*). Greater compression of the hindlimb joints is thought to contribute to greater springiness and impulsion of horses with high gait scores (*Holmstrom & Drevemo, 1997*). Similarly, increased fetlock extension is a result of greater applied vertical load on the limbs (*Merkens & Schamhardt, 1994*; *McGuigan & Wilson, 2003*), which will increase dorsoventral displacement (*Biau & Barrey, 2004*; *Hobbs & Clayton, 2013*) and potentially result in a longer aerial phase between diagonal contacts. Differences in force production between Warmbloods and Lusitanos have been reported in collected trot, with Warmbloods producing higher vertical impulses in all limbs (*Clayton, Schamhardt & Hobbs, 2017*). Regardless of whether they are lame or sound, Quarter Horses produce lower mass-normalized ground reaction forces than Warmbloods (*Back et al., 2007*). Kinematic suitability for dressage has been compared between different breeds (*Barrey et al., 2002*), as such, horses may be selected for their ability to produce higher forces at the ground, which will increase dorsoventral displacement and therefore give the impression of greater 'elevation'.

For the rider, gaits with suspension phases require pelvic mobility and control in order to follow and amplify the horse's motion (*Münz, Eckardt & Witte, 2014*; *Byström et al., 2015*; *Engell et al., 2016*). In skilled riders, the pelvis rotates from anterior to posterior tilt over the stride cycle with a smaller amount of lateral tilt (*Münz, Eckardt & Witte, 2014*), whilst the trunk maintains a more consistent vertical posture and the head a more consistent and stiller horizontal posture (*Eckardt & Witte, 2016*). The posture of the pelvis and upper body segments dictates how pressure is distributed under the saddle (*De Cocq et al., 2009*; *Gunst et al., 2019*), which affects the aids communicated to the horse and also impacts on the horses' balance (*De Cocq et al., 2010b*). In skilled riders, pelvic motion is independent of trunk, head or other segment motion, which requires dynamic postural control (*Engell et al., 2016*). When the rider achieves an advanced level of dynamic postural control, it improves the harmony between horse and rider (*Peham et al., 2001*; *Münz, Eckardt & Witte, 2014*), and translates to higher average dressage scores (*Peham et al., 2001*). Skilled riders control body position by coordinating activity level and antagonistic timing of Erector Spinae and Rectus Abdominis muscles (*Terada, 2000*; *Pantall, Barton & Collins, 2009*), whilst novice riders display energetically inefficient co-activation of Erector Spinae and Rectus Abdominis muscles (*Pantall, Barton & Collins, 2009*) and use Adductor Magnus to stabilize the trunk (*Terada, 2000*). Phasic activity in Rectus Abdominis in mid-stance is used to stabilize the rider's trunk and enable the rider to follow the horse's movement by rotating the pelvis posteriorly as the horse's body reverses direction from downward to upward motion (*Terada et al., 2004*; *Pantall, Barton & Collins, 2009*). In addition, phasic activity of the upper and middle Trapezius in early stance is used to stabilize the head, neck and scapula during impact of the diagonal limbs (*Terada et al., 2004*).

As suggested previously, a number of attributes of the horse that will influence dressage scores relate to 'elevation'. The challenge for the rider is therefore often associated with their ability to maintain dynamic postural control and harmony with the horse whilst coping with the large vertical and longitudinal accelerations and decelerations of the horse's trunk in trot (*Terada, 2000*; *Byström et al., 2015*). Skilled riders are said to have a stabilizing effect on the horse, as shown by a reduction in motion pattern variability

(*Peham et al., 2004*). Improved rider-horse harmony will reduce disruption in temporal variables and dorsoventral motion associated with horse performance (*Wolframm, Bosga & Meulenbroek, 2013*). Again, pelvic motion from anterior to posterior tilt of the rider was found to significantly increase nose up trunk tilt of the horse during trotting (*Münz, Eckardt & Witte, 2014*). Variations in rider pelvic posture are reported (*Byström et al., 2009*; *Münz et al., 2013*; *Eckardt, Münz & Witte, 2014*; *Münz, Eckardt & Witte, 2014*; *Alexander et al., 2015*; *Byström et al., 2015*; *Eckardt & Witte, 2016*; *Engell et al., 2016*), but also depend on the goal of the rider. When actively influencing the horse to improve collection in trot, skilled riders have greater posterior pelvic tilt throughout the stride (*Byström et al., 2015*; *Engell et al., 2016*).

Less knee flexion-extension ROM in skilled riders may also relate to the rider's ability to cope with the motion of the horse, with less of a tendency in skilled riders to pull up the knees in an effort to remain balanced (*Byström et al., 2015*). A stiller leg will improve the rider's ability to provide consistent and precise aids to the horse, resulting in more finite speed, gait and/or movement changes. The ability to maintain consistent contact with the bit at all gaits is also necessary to facilitate good rider-horse communication (*Eisersiö et al., 2013*; *Von Borstel & Glißman, 2014*). Perturbations due to the motion of the horse's trunk are accommodated by the rider with the apparent goal of allowing the rider's hand to maintain a consistent position relative to the bit (*Terada, Clayton & Kato, 2006*; *Eisersiö et al., 2013*). *Eckardt & Witte (2016)* reported an increase in flexion-extension ROM of the elbow in skilled riders to effect this. *Terada, Clayton & Kato (2006)* showed that pitching rotations of the rider's trunk were compensated by coordinated flexion-extension of the shoulder and elbow joints so the distance from the rider's wrist to the bit changed by no more than 1.5 cm. These movements were controlled by activation of Biceps Brachii in early stance and Triceps Brachii in late stance (*Terada, 2000*).

## Canter

The canter is an asymmetrical gait with three beats in the sequence (1) trailing hindlimb, (2) leading hindlimb and trailing forelimb together then (3) leading forelimb. Lift off of the trailing forelimb is followed by a period of suspension (*Clayton, 1994b*). A good quality canter is associated with greater stride regularity, increased dorsoventral displacement and activity, increased longitudinal activity and increased vector of propulsion (*Biau & Barrey, 2004*). For extended canter the following additional attributes have also been identified; shorter trailing hindlimb contact duration, longer stride length, faster velocity, decreased step length between forelimbs, increased step length between hindlimbs and a longer step length between leading hindlimb and trailing forelimb (*Deuel & Park, 1990a*).

Canter is performed at a faster speed than its equivalent trot (for example, extended trot: $4.93 \pm 0.14$ ms$^{-1}$ (*Clayton, 1994a*; *Clayton, 1994b*); extended canter: $7.03 \pm 0.07$ ms$^{-1}$ (*Deuel & Park, 1990a*)), and due to the limb sequencing pattern has a larger range of trunk pitching than walk or trot (*Dunbar et al., 2008*). As such, heightened pelvic mobility and postural control are required by the rider to maintain balance and harmony with the horse (*Olivier et al., 2017*), although greater synchronicity is possible due to canter being a three-beat gait (*Wolframm, Bosga & Meulenbroek, 2013*). In skilled riders, pelvic

anterior-posterior ROM (*Münz, Eckardt & Witte, 2014*) trunk lateral bending ROM and left knee flexion-extension ROM (*Eckardt & Witte, 2016*) are smaller compared to less skilled riders. As trunk ROM of the horse in pitch and longitudinal forces increase, a closer coupling of the pelvis in anterior-posterior tilt and greater control of the upper body are required in order to follow the phasic motions of the horse (*Lovett, Hodson-Tole & Nankervis, 2005*; *Wolframm, Bosga & Meulenbroek, 2013*; *Münz, Eckardt & Witte, 2014*). A reduction in rider trunk lateral bending ROM is likely to reduce amplification of the asymmetry of the gait and improve medio-lateral and rotational stability in the horse (*Symes & Ellis, 2009*). As with trot, a stiller left leg is likely to improve communication with the horse and probably reflects better rider balance (*Byström et al., 2015*; *Eckardt & Witte, 2016*), whereas a more mobile left leg may disrupt canter with a right lead more than a left lead (*Symes & Ellis, 2009*).

## Other gaits and movements

Other studies that have focussed on horse performance include information on the quality of transitions, canter lead changes and the artificial diagonal gaits of passage and piaffe that are performed only at the highest levels of competition.

Transitions performed in dressage differ from transitions performed naturally, as the trigger is a learned cue from the rider. Ideally, the gait before and after the transition is performed at steady state, without changing speed or stride rate (*Tans, Nauwelaerts & Clayton, 2009*). However, the stride before the transition may need to change to accommodate a difference in speed between the two gaits, especially in downward transitions that cross more than one gait as in canter-walk or canter-halt (*Biau, Lemaire & Barrey, 2002*). The use of a pre-transition cue or half-halt by the rider to balance the horse will improve the quality of the transition (*Byström et al., 2015*). Under these conditions, mechanical or metabolic stimuli that drive a natural transition are overridden by the trained response (*Tans, Nauwelaerts & Clayton, 2009*). A variety of upward and downward transitions have been studied. Clean transitions between walk and trot with no intermediate steps are a feature of better performance in elite dressage horses (*Argue & Clayton, 1993b*). For trot-halt and halt-trot, longer suspension duration, hind first dissociation and lift off and smaller duty factors are associated with better quality (*Tans, Nauwelaerts & Clayton, 2009*). Superior downward transitions also include longer transition durations, increased dorsoventral activity, but with decreased energy and frequency variation (*Biau, Lemaire & Barrey, 2002*).

As transitions are effected by the rider in dressage, the ability of the rider to communicate well with the horse is essential. Aids are given by altering pressure through the reins, legs and/or seat (*De Cocq et al., 2010b*; *Kuhnke et al., 2010*; *Von Borstel & Glißman, 2014*; *Egenvall et al., 2015*). The quality and consistency of rein tension is, either directly or indirectly, an important factor in judging transitions, with lighter, more consistent and symmetrical rein tension considered to be more desirable (*Kuhnke et al., 2010*; *Von Borstel & Glißman, 2014*). The position of the centre of pressure (COP) under the saddle influences the direction of travel of the horse and symmetric or asymmetric leg aids are used to indicate changes in gaits and movements (*De Cocq et al., 2010b*). The frequency and magnitude of

leg pressure varies between riders and, as of yet, better-quality communication with the horse has not been defined (*De Cocq et al., 2010b*).

As with transitions, canter lead changes are triggered by a learned cue from the rider, so precise communication between rider and horse is necessary to execute the movements correctly. They may be one-time changes, that is where the canter lead changes with each stride, or the changes may be made every second, third or fourth stride. Superior canter lead changes have attributes that are similar to canter itself, including greater suspension duration, shorter hindlimb tripedal contact duration, decreased step length between forelimbs, and a longer step length between leading hindlimb and trailing forelimb (*Deuel & Park, 1990a*). In addition, canter lead changes judged to be better quality include a shorter hindlimb and longer forelimb swing duration, a shorter trailing forelimb contact duration and a longer airborne step (*Deuel & Park, 1990a*). To date, the authors are not aware of any scientific studies that have investigated the effect of rider skill or the demands on the rider when performing canter lead changes.

Passage and piaffe are diagonal gaits that require higher levels of collection; they are performed at slow speed and with little or no forward movement of the horse (*Clayton, 1997*; *Clayton & Hobbs, 2017b*). In passage, gait quality is associated with hind-first dissociation and a shorter hindlimb stance duration (*Clayton, 1997*). These qualities result in a greater nose up trunk posture, which positions the horse's COP more towards the hindlimbs. This gives the impression of, and results in, greater weight bearing on the hindlimbs (*Clayton, Schamhardt & Hobbs, 2017*). Passage has large vertical impulses that increase the vertical excursion of the horse compared to collected trot (*Weishaupt et al., 2009*; *Clayton, Schamhardt & Hobbs, 2017*). Less is known about piaffe. From what is known, the footfall sequence is reported to be highly variable between horses, but still features the same attributes of gait quality as passage described above (*Clayton, 1997*). Compared with scores awarded in 1992, in 2008 dressage scores in top level competition were higher in horses with their head posture behind the vertical in piaffe, although this may have reflected the use of a specific training technique that was popular at that time (*Lashley et al., 2014*). Posture of the rider is reported to change with increasing collection of the horse, such as in passage and piaffe. When giving an aid to collect, the rider's pelvis rotates posteriorly and the trunk rotates anteriorly, thereby flexing the lumbar spine (*Byström et al., 2015*). Possibly due to the greater vertical excursion of both the horse and rider, pelvic rotation and displacement are more closely coupled to the horse (*Byström et al., 2015*). Further studies are required to define skill when performing these movements.

## Limitations

The search strategy for this review was designed to capture all of the current scientific evidence relating directly or indirectly to determinants of performance in dressage. However, it is evident from the manual addition of 94 articles that the search did not capture all of the available literature.

A small number of studies were included in the theoretical model for the rider, as these were the only studies reporting significant findings, separated by gait, that were associated with superior rider performance.

Since dressage is judged subjectively and multiple criteria are considered in relation to scoring each movement, it is to be expected that there will be variability in the scores awarded by different judges. Some judges show greater variability (*Stachurska & Bartyzel, 2011*) and others award higher scores on a nationalistic basis (*Deuel, 1989*).

Finally, this review considered only determinants of performance of the horse and rider that may be observed at a particular moment in time. The review did not consider the training, welfare and/or health, amongst other factors, that may have contributed to that performance.

## CONCLUSION

This review has identified objective measurements of horse performance that are associated with higher scores in dressage. The articles included in the review and additional source information were then used to develop a theoretical model to link the characteristics displayed by skilled riders to horse performance themes. From this model it could be concluded that the posture and ROM of the rider's pelvis, trunk, knee and head and, importantly, the timing of rider pelvic and trunk motion in relation to the movement of the horse are likely to influence temporal, trunk motion and impulsion variables in the horse. The information from this model will be used to develop an empirical study to test the relative strength of association between impairment and performance in able-bodied and Para Dressage riders.

## ACKNOWLEDGEMENTS

The authors would like to acknowledge Professor Andy Clegg (University of Central Lancashire) for his advice and support and Elizabeth Littlefair (formerly University of Central Lancashire) for her contribution to the screening and data extraction process.

### Funding

This work was supported by the Fédération Equestre Internationale (FEI Para-Equestrian sport: 2018–2021). The funders had no role in study design, data collection and analysis, decision to publish, or preparation of the manuscript.

### Grant Disclosures

The following grant information was disclosed by the authors:
Fédération Equestre Internationale: FEI Para-Equestrian sport: 2018–2021.

### Competing Interests

Hilary M. Clayton is the CEO of Sport Horse Science.

### Author Contributions

- Sarah Jane Hobbs and Lindsay St George conceived and designed the experiments, performed the experiments, analyzed the data, prepared figures and/or tables, authored or reviewed drafts of the paper, and approved the final draft.

- Janet Reed performed the experiments, analyzed the data, prepared figures and/or tables, and approved the final draft.
- Rachel Stockley, Clare Thetford, Jonathan Sinclair, Jane Williams and Kathryn Nankervis conceived and designed the experiments, authored or reviewed drafts of the paper, and approved the final draft.
- Hilary M. Clayton conceived and designed the experiments, analyzed the data, authored or reviewed drafts of the paper, and approved the final draft.

## Data Availability

The systematic review methods and comprehensive results are available in the Supplemental Files.

## Supplemental Information

Supplemental information for this article can be found online at http://dx.doi.org/10.7717/peerj.9022#supplemental-information.

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
