# Peer review of "A scoping review of determinants of performance in dressage"

_PeerJ, doi:10.7717/peerj.9022_

## Round 0.1 · original submission · Minor Revisions

Please carefully note Reviewer 1's recommendations and address them succinctly. Please consider whether or not your paper is scoping review or systematic review as per reviewer number two's concerns and address accordingly.

Reviewer 1 ·

Basic reporting

Generally a very thoughtful review that has high relevance for equine researchers and practical use for the field of equestrian dressage in practice. The literature has been well selected for the purpose of the review, which was to identify quantitative measurements for dressage performance for both the horse and/or the rider. This review was very well written, and the text flowed logically.

To further improve the reading of the manuscript, I would recommend to start the first sentences of the introduction with a reference to para dressage or dressage, similarly to the abstract, as the first paragraph of the introduction takes quite long to reach the topic of the review, (Para) Dressage.

Experimental design

This was a well-designed systematic literature review.


I am missing the aspect of equine welfare, and rider skills in relation to welfare. Specific riding methods, such as Rollkur, which are mentioned in your keywords, and their effect on kinematic variables and/or welfare of the horse fail to appear in the discussion. It could be argued that horses whose welfare is impacted have potentially lower performance. Was there a particular reason why this aspect did not appear more in the discussion?

Validity of the findings

Based on my knowledge, all reported findings and conclusions are valid. I would like to see in the limitation section a discussion of the limitation of performance scores as a reference for performance, given the sometimes low agreement between judge scores (e.g. Stachurska and Bartyzel, 2011).

I have a few specific questions about statements:

Line 238-241: This is very interesting! However, I am missing the link between energetic efficiency and performance. It would also likely be more efficient for some horses to transition to a faster gait instead of lengthening the stride, but adapting stride length and frequency irrespective of energetic efficiency is a main requirement for dressage. Could you please expand on the aspects of efficiency, regularity and the vectors of braking and propulsion found to be relevant to dressage by Biau and Barrey (2004)?

Line 267-269: Is the step frequency at the walk necessarily rapid? Does it not also depend on the ability of the horse and rider to increase stride length and decrease stride frequency?

Line 302: what do you mean with “tracking length”? do you mean “over-tracking distance”? It would be nicer to have consistency in the terms used, especially as you use “over-tracking” in the following sentence.

Line 303: “longer strides are achieved by increasing propulsion and the propulsion vector…” Is the propulsion coming from the hind limb? In what direction is the propulsion vector going? Please further explain this.

Line 314-315: “these traits are regarded positively by judges”. Please provide references for this statement or combine the sentence with the previously cited results from Biau and Barrey (2004).

Line 324: Why and how? This statement should be supported by references.

Line 340: “creating impulsion”. Do you mean “impulsion” in the equestrian sense or in the biomechanical sense? It could be helpful to define this more thoroughly, maybe in lines 290-292, where the term first appears.

Additional comments

I have some further comments on the manuscript:

Specific comments:
Abstract:
Line 38: is there a difference between “posture” and “postural position”? “Postural position” seems to me like a word redundance. Also on line 357.

Introduction:
Line 50: it would be helpful to start the first sentence with a reference to Para Dressage, and then go on to explain how Paralympic disciplines are defined. The paragraph takes quite long to reach the topic of the review, (Para) Dressage. It is much nicer in the beginning of the abstract (line 14).
Line 76: for walk, you reference only Clayton 1995. Back et al. 1996 also analysed the walk in relation to gait quality. This reference may be added.
Lines 90-103: this is where the welfare aspect of rider skills could be discussed.

Survey methodology:
Line 155: included instead of include.
Line 156: selected instead of included?
Line 159, 160: Remove space between the backslash and the second word.
Line 160: what is “adjustability”? This term has not been previously described and is probably not common. Please define the term.
Line 160: same comment for “connection”. Please define the term.
Line 174: please remove space between the backslash and the second word.

Discussion:
Line 219: “to improve their performance” instead of “to develop an improved performance”.
Line 219-222: This statement needs one or several references.
Line 348: “Lusitanos” instead of “Lusitano’s”
Line 347-249: These breed specific differences are interesting. It would be nice to add a few references on breed-specific differences in locomotion (e.g. Barrey et al. 2002).
Line 357: “postural position” like a word redundance. Would “posture” not be enough?
Line 443-452: very nice!

Table S1: the expression is "throughness", from the German “Durchlässigkeit”, and not "thoroughness".

Reviewer 2 ·

Basic reporting

See below

Experimental design

See below

Validity of the findings

See below

Additional comments

General comments
The manuscript presents a review of scientific evidence related to dressage performance. The text is well written, with very few typographical errors.
The aim of providing a scientific background for classification of disability levels in para-dressage is legit, but this goal is little touched upon in the discussion. I started my reading with the search strategy, and read through to the end of discussion before I returned to the introduction, and I was rather surprised when I read the incentive for the study. I also wondered whether this review of the literature was asked for by the Paralympic committee or if this was an initiative by the authors themselves.
The review adds moderately to the existing body of literature. Similar material is available in book chapters and previous reviews mentioned in the current manuscript, though all of this information is not available in the same paper, as the authors point out.

The study is reported as a systematic review, but I would argue that what has been undertaken here is not a systematic review, rather it’s a scoping review. A systematic review summarizes multiple studies that have addressed the same research question, while a scoping review maps the existing literature relevant for a particular question or area of interest. It is obvious already from the aims, that this study addresses a broader area rather than a single focused question and the results section does not present evidence to inform practice, rather it’s presents mapping of the literature available and overview of study findings. This is not suggesting a lower value of the publication, but I feel that it’s important to present it in the correct framework, and this review also does not benefit from trying to squeeze itself into the systematic review framework where it clearly doesn’t fit. The guidelines of this journal refers to the following publication for guidance whether a review is a systematic review or a scoping review.

Munn Z, Peters MDJ, Stern C, Tufanaru C, McArthur A, Aromataris E. 2018. Systematic review or scoping review? Guidance for authors when choosing between a systematic or scoping review approach. BMC Medical Research Methodology 18:143 DOI: 10.1186/s12874-018-0611-x.

I suggest that you read this publication. Related to this issue, use of the PICO formula does not fit a scoping review, this search strategy is mainly tailored towards systematic review of intervention studies and in your form you state intervention was non-applicable in your case.

Materials and methods
The initial search strategy appears non-optimal, the chosen combination of words could pose a risk that relevant papers are missed. However, I interpret that you extended the search using keywords that appeared relevant when you went through your search results. Is that a correct interpretation? It is difficult to follow the search development from the supplement. I am not sure what the numbers listed actually mean, it seems it’s the number of hits for each word. I think some explanatory notes would be good to make this more interpretable. Also, why is the VetMed sheet empty?
I miss the prisma checklist, which is mandatory for systematic reviews according to the journal guidelines. But as I understand this is not mandatory for scoping review.
Your inclusion criteria seem to suggest that every paper needs to contain something on both horse and rider. Is that really the case? These criteria are also not easy to understand. I suggest that you include an example after each except the first one. For number two you can for example write e.g. optic motion capture/ kinematic data.
It would be good if you state when the search was performed, even though this information is given implicitly by the time span of the search.

Results
Please add journal, volume and page numbers to your list of included papers if feasible. This will make the supplements more valuable resources for readers.

For me, figure 2 is difficult to read, there is a lot of arrows crossing. Are the relationships illustrated in figure 2 based on findings in the reviewed papers or suggestions by authors of these papers, and/or your own reasoning? Please be clear about this in the text and figure legend. Regardless, I think these interrelationships are open questions, apart from the few that we have data on. For example, in the text you say that ’stiller leg will improve the rider’s ability to provide consistent and precise aids to the horse, resulting in more finite speed, gait and/or movement changes’ in trot while the figure suggests that less knee flexion-extension does not have an influence on joint kinematics.

Discussion:
I feel the discussion could be more condensed and be more concluding and interpreting rather than just reporting. There is no clear connection or reference to figure 2. I don’t mean that you have the rewrite the whole discussion, but I suggest that you go over the text with this in mind.

Line 245: please add that the comparison between a smaller and a taller horse is only applicable if they are walking at the same speed. Unless I misunderstood this in someway.

Line 267: please add a reference for rider pelvis pitch rotation.

Line 268: I’m not sure if I agree with this. Both walk and trot have two cycles of rider pitch per stride, and in trot the stride frequency is higher. Further, depending on the speed of the respective gaits, walk step frequency is not that much higher compared to trot.

The conclusions are comprehensive and well written, and seem relevant. But the discussion lacks connection to the conclusions. If you go through the discussion according to my suggestion above, I hope this will improve.

Supplement
One experimental study is classified as before and after. According to my understanding before and after studies are non-experimental. If you have a different understanding please provide (me) a reference.

---

## Round 0.2 · Minor Revisions

Please see the very minor and short revision opportunities from the reviewers. Thank you for your patience.

Reviewer 1 ·

Basic reporting

As before, the clarity of language was excellent.

Experimental design

The authors have adapted their description of the review (scoping instead of systematic) according to reviewer two's comments. The comments I had personally on the aspect of equine welfare were addressed in the discussion/conclusion.

Validity of the findings

All my comments were addressed in the revised edition of this manuscript. Thank you.

Additional comments

I just have two small comments:

on line 322: there is a space missing between "horses" and the reference.

on line 370: the reference is not complete "(Barrey et al.,"

Reviewer 2 ·

Basic reporting

Consider if you want to add to the introduction that the review was conducted part of, or in preparation for, a larger study commissioned by the FEI. This makes it more understandable why the para dressage perspective, thinking that I will not be the only one wondering about this. I can’t find any statement about source of funding, but maybe this is provided to the journal separately.

Experimental design

I am happy that the authors appreciated my comments regarding the paper being a scoping review. I think the paper and the supplementary material benefit from the changes made, adapting them to the scoping review framework.
The authors have amended the unclear points regarding the search strategy in a good way.

Validity of the findings

I appreciate that the authors have provided a better explanation for figure former 2, now 3. I feel the supplement linking the figure and the discussion makes it more transparent, which I felt was an important issue in the previous submission. Only, you need to change figure number in the supplement, it is referring to figure 2 (should be 3). I’m still not a huge fan of figures 2-3, but a particular graphical illustration will seldom appeal to all readers.

Supplementary text S2: ‘Systematic Review and/or Meta-Analysis Rationale’ is no longer relevant, I assume this was left in by accident and is to be removed?

I’m fine with the amendments that have been made to the discussion. I’m less bothered by it being lengthy now when re-reading it from a scoping review perspective only and not searching for the type of conclusive remarks and recommendation statements that should be part of a systematic review.

---

## Round 0.3 · accepted · Accept

Thank you for your patience and professionalism throughout the review process.